# Natural Frequencies Calculation of Composite Annular Circular Plates with Variable Thickness Using the Spline Method

Saira Javed 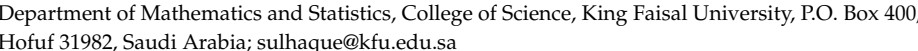

Department of Mathematics and Statistics, College of Science, King Faisal University, P.O. Box 400, Hofuf 31982, Saudi Arabia; sulhaque@kfu.edu.sa

**Abstract:** The present study adds to the knowledge of the free vibration of antisymmetric angle-ply annular circular plates with variable thickness for simply supported boundary conditions. The differential equations in terms of displacement and rotational functions are approximated using cubic spline approximation. A generalized eigenvalue problem is obtained and solved numerically for an eigenfrequency parameter and an associated eigenvector of spline coefficients. The vibration of the annular circular plates is examined for circumferential node number, radii ratio, different thickness variations, number of layers, stacking sequences and lamination materials.

**Keywords:** free vibration; antisymmetric; shear deformation; spline approximation; eigenfrequency parameter

## 1. Introduction

Many engineering structures consist of composite annular circular plates of variable thickness. The distinguishing features of variable thickness plates are to obtain desired frequency and to reduce weight, size and cost of the structure. The required properties of the structure can be attained by selecting the best aspect of the constituent layers in terms of choice of materials, number of layups, thickness variation of each layer and boundary conditions.

Most of the researchers worked on functionally graded annular plates [1–6]. Whereas Ref. [7] studied the free and forced vibration of annular plates made of carbon fiber-carbon nanotube-polymer hybrid composites, Ref. [8] investigated the free vibration of an annular mirror for the optical aberration representation. A number of methods were used for analyzing vibration of annular plates; among them, the Ritz method was used by [9], the Fourier series by [10], the Chebyshev collocation technique by [11], the Jacobi–Ritz method by [12] and Carrera's unified formulation by [13]. The FEM method was used to study the free and static vibration of FGM plates by [14]. However, the present research uses spline approximation to analyze composite plates of varying thickness with each layer consisting of different material. Cross-ply laminates were studied by [15]. Higher-order shear deformation theories with a unified model were studied for composite plates by [16]. In Ref. [17], the free vibration of eccentric annular plates was analyzed. In Ref. [18], annular plates coupled with fluids were investigated. The vibration of annular sector plates was examined in [19]. Some researchers, such as those in [20], used a B-spline to analyze rectangular plates using classical thin plate theory, whereas the current research uses spline approximation using first-order shear deformation theory and composite plates having variable thickness were used. Moreover, Ref. [21] used spline approximation to investigate nonhomogeneous circular plates of quadratic thickness, but, in this paper, composite annular circular plates of linear, exponential and sinusoidal thickness variations were considered. Ref. [22] used spline approximation to analyze rectangular orthotropic plates of variable thickness using classical and refined theories, whereas this research uses first-order shear deformation theory to examine the vibration of antisymmetric angle-ply annular circular plates of varying thickness using first order shear deformation theory.

The novelty of the current research is that antisymmetric angle-ply plates of linear, exponential and sinusoidal thickness variations have not been used by any of above-mentioned researchers to analyze free vibration of annular circular plates. Moreover, each layer being of variable thickness and consisting of different material is also a novelty of the current research. The displacement and rotational functions are approximated using the cubic spline method. The problem is solved using the eigen solution technique to obtain the frequency parameters. Obtained results are presented by graphs and tables.

## 2. Formulation of the Problem

Consider a composite laminated annular circular plate with an arbitrary number of layers, as shown in Figure 1. Consider $r_a = a$ is the inner radius and $r_b = b$ is the outer radius of the annular circular plate and $\ell = b - a$ is the width of the annular circular plate. The curvilinear coordinate system $(r, \theta, z)$ is fixed at its reference surface, which is taken to be its middle surface.

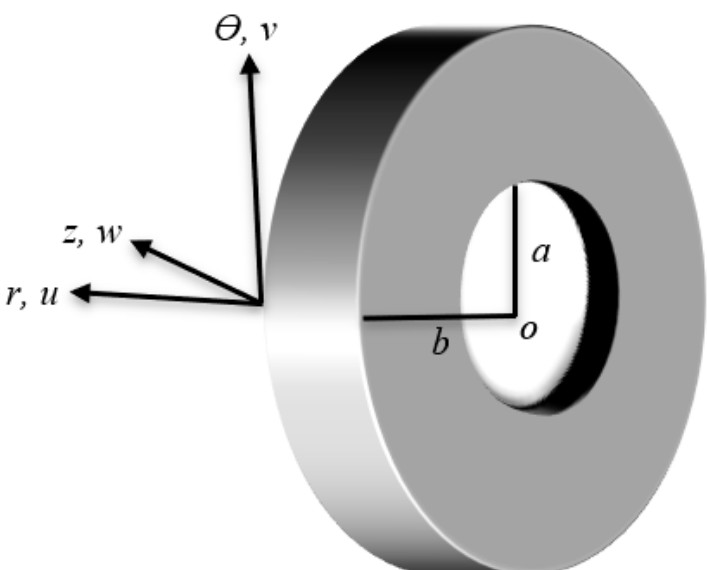

**Figure 1.** Geometry of annular circular plate.

According to [23], the displacement components are assumed to be

$$u(r, \theta, z, t) = u_0(r, \theta, t) + z\, \psi_r(r, \theta, t)$$
$$v(r, \theta, z, t) = v_0(r, \theta, t) + z\, \psi_\theta(r, \theta, t) \qquad (1)$$
$$w(r, \theta, z, t) = w_0(r, \theta, t)$$

The equations of stress-resultants and displacements are of the form

$$
\begin{pmatrix} N_r \\ N_\theta \\ N_{r\theta} \\ M_r \\ M_\theta \\ M_{r\theta} \end{pmatrix} =
\begin{pmatrix}
A_{11} & A_{12} & A_{16} & 0 & 0 & 0 \\
A_{12} & A_{22} & A_{26} & 0 & 0 & 0 \\
A_{16} & A_{26} & A_{66} & 0 & 0 & 0 \\
0 & 0 & 0 & D_{11} & D_{12} & D_{16} \\
0 & 0 & 0 & D_{12} & D_{22} & D_{26} \\
0 & 0 & 0 & D_{16} & D_{26} & D_{66}
\end{pmatrix}
\begin{pmatrix} \varepsilon_r \\ \varepsilon_\theta \\ \gamma_{r\theta} \\ \kappa_r \\ \kappa_\theta \\ \kappa_{r\theta} \end{pmatrix}
\quad
\begin{pmatrix} Q_\theta \\ Q_r \end{pmatrix} = K
\begin{pmatrix} A_{44} & A_{45} \\ A_{45} & A_{55} \end{pmatrix}
\begin{pmatrix} \gamma_{\theta z} \\ \gamma_{rz} \end{pmatrix}
\qquad (2)
$$

where

$$\varepsilon_r = \frac{\partial u_0}{\partial r} + z\frac{\partial \psi_r}{\partial r}, \; \varepsilon_\theta = \frac{1}{r}\frac{\partial v_0}{\partial \theta} + \frac{u_0}{r} + z\left(\frac{1}{r}\frac{\partial \psi_\theta}{\partial \theta} + \frac{\psi_r}{r}\right),$$
$$\gamma_{r\theta} = \frac{1}{r}\frac{\partial u_0}{\partial \theta} + \frac{\partial v_0}{\partial r} - \frac{v_0}{r} + z\left(\frac{1}{r}\frac{\partial \psi_r}{\partial \theta} + \frac{\partial \psi_\theta}{\partial r} - \frac{\psi_\theta}{r}\right),$$
$$\kappa_r = \frac{\partial \psi_r}{\partial r}, \quad \kappa_\theta = \frac{1}{r}\frac{\partial \psi_\theta}{\partial \theta} + \frac{\psi_r}{r}, \quad \kappa_{r\theta} = \frac{\partial \psi_\theta}{\partial r} + \frac{1}{r}\frac{\partial \psi_r}{\partial \theta} - \frac{\psi_\theta}{r},$$
$$\gamma_{rz} = \psi_r + \frac{\partial w}{\partial r} \text{ and } \gamma_{\theta z} = \psi_\theta + \frac{1}{r}\frac{\partial w}{\partial \theta} \tag{3}$$

The thickness variation of the $k^{th}$ layer of the plate is assumed in the form which can be seen in Figure 2:

$$h_k(r) = h_{0k}\, g(r) \tag{4}$$

where $g(r) = 1 + C_\ell\left(\frac{r-r_a}{\ell}\right) + C_e \exp\left(\frac{r-r_a}{\ell}\right) + C_s \sin\left(\pi\left(\frac{r-r_a}{\ell}\right)\right)$ and $h_{0k}$ is a constant thickness of the $k^{th}$ layer. The thickness of the plate becomes uniform when $g(r) = 1$.

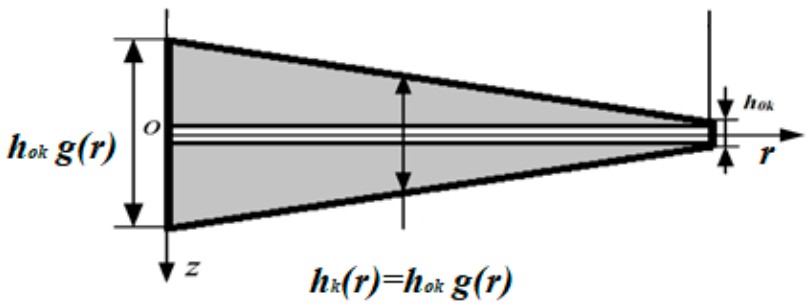

**Figure 2.** Plate of variable thickness.

Since the thickness is assumed to be varying along the radial direction, one can define the elastic coefficients $A_{ij}$, $B_{ij}$ and $D_{ij}$ (extensional and bending-extensional coupling and bending stiffness), corresponding to layers of uniform thickness with superscript '$c$', by

$$A_{ij} = A_{ij}^c\, g(r), \; B_{ij} = B_{ij}^c\, g(r), \; D_{ij} = D_{ij}^c\, g(r) \tag{5}$$

$$A_{ij}^c = \sum_k \overline{Q}_{ij}^{(k)}(z_k - z_{k-1}), \; B_{ij}^c = \frac{1}{2}\sum_k \overline{Q}_{ij}^{(k)}(z_k^2 - z_{k-1}^2), D_{ij}^c = \frac{1}{3}\sum_k \overline{Q}_{ij}^{(k)}(z_k^3 - z_{k-1}^3) \quad \text{for } i,j = 1,2,6 \tag{6}$$

and

$$A_{ij}^* = K\sum_k \overline{Q}_{ij}^{(k)}(z_k - z_{k-1}) \text{ for } i,j = 4,5 \tag{7}$$

where $K$ is the shear correction coefficient and $z_{k-1}$ and $z_k$ are the boundaries of the $k$-th layer. The value of $K$ for a general laminate depends on lamina properties and lamination scheme.

The displacement components $u_0$, $v_0$ and $w$ and shear rotations $\psi_r$ and $\psi_\theta$ are assumed in the form

$$u_0(r,\theta,t) = U(r)\, e^{n\theta} e^{i\omega t},$$
$$v_0(r,\theta,t) = V(r)\, e^{n\theta}\, e^{i\omega t},$$
$$w(r,\theta,t) = W(r)\, e^{n\theta}\, e^{i\omega t}, \tag{8}$$
$$\psi_r(r,\theta,t) = \Psi_r(r)\, e^{n\theta}\, e^{i\omega t},$$
$$\psi_\theta(r,\theta,t) = \Psi_\theta(r)\, e^{n\theta}\, e^{i\omega t},$$

where $r$ and $\theta$ are polar coordinates which describe the radial and circumferential direction, $\omega$ is the angular frequency of vibration, $t$ is the time and $n$ is the circumferential node number.

Substituting Equation (8) into the equilibrium equation and replacing the coefficients $A_{16}$, $A_{26}$, $A_{45}$, $B_{11}$, $B_{12}$, $B_{22}$, $B_{66}$, $D_{16}$ and $D_{26}$ identically with zero for antisymmetric angle-ply laminates, the resulting equation becomes, in the matrix form,

$$
\begin{bmatrix}
L_{11} & L_{12} & L_{13} & L_{14} & L_{15} \\
L_{21} & L_{22} & L_{23} & L_{24} & L_{25} \\
L_{31} & L_{32} & L_{33} & L_{34} & L_{35} \\
L_{41} & L_{42} & L_{43} & L_{44} & L_{45} \\
L_{51} & L_{52} & L_{53} & L_{54} & L_{55}
\end{bmatrix}
\begin{Bmatrix}
U \\ V \\ W \\ \psi_R \\ \psi_\Theta
\end{Bmatrix}
=
\begin{Bmatrix}
0 \\ 0 \\ 0 \\ 0 \\ 0
\end{Bmatrix}
\tag{9}
$$

where

$$L_{11} = A_{11}g\frac{d^2}{dr^2} + A_{11}g'\frac{d}{dr} + + A_{11}g\frac{1}{r}\frac{d}{dr} + A_{12}g'\frac{1}{r} - A_{22}g\frac{1}{r^2} + A_{66}g\frac{n^2}{r^2} + I_1\omega^2$$

$$L_{12} = A_{12}g\frac{n}{r}\frac{d}{dr} + A_{66}g\frac{n}{r}\frac{d}{dr} + A_{12}g'\frac{n}{r} - A_{22}g\frac{n}{r^2} - A_{66}g\frac{n}{r^2}, \; L_{13} = 0$$

$$L_{14} = 2B_{16}g\frac{n}{r}\frac{d}{dr} + B_{16}g'\frac{n}{r}$$

$$L_{15} = B_{16}g\frac{d^2}{dr^2} + B_{16}g'\frac{d}{dr} - B_{26}g\frac{1}{r}\frac{d}{dr} + B_{26}g\frac{1}{r^2} + B_{26}g\frac{n^2}{r^2} - B_{16}g'\frac{1}{r}$$

$$L_{21} = A_{12}g\frac{n}{r}\frac{d}{dr} + A_{66}g\frac{n}{r}\frac{d}{dr} + A_{22}g\frac{n}{r^2} + A_{66}g'\frac{n}{r} + A_{66}g\frac{n}{r^2}$$

$$L_{22} = A_{66}g\frac{d^2}{dr^2} + A_{66}g'\frac{d}{dr} + A_{66}g\frac{1}{r}\frac{d}{dr} + A_{22}g\frac{n^2}{r^2} - A_{66}g'\frac{1}{r} - A_{66}g\frac{1}{r^2} + I_1\omega^2$$

$$L_{23} = 0$$

$$L_{24} = B_{16}g\frac{d^2}{dr^2} + B_{16}g'\frac{d}{dr} + 2B_{16}g\frac{1}{r}\frac{d}{dr} + B_{26}g\frac{1}{r}\frac{d}{dr} + B_{26}g'\frac{1}{r} + B_{26}g\frac{1}{r^2} + B_{26}g\frac{n^2}{r^2}$$

$$L_{25} = 2B_{26}g\frac{n}{r}\frac{d}{dr} + B_{26}g'\frac{n}{r}, \; L_{31} = 0, \; L_{32} = 0$$

$$L_{33} = KA_{55}g\frac{d^2}{dr^2} + KA_{55}g'\frac{d}{dr} + KA_{55}g\frac{1}{r}\frac{d}{dr} + KA_{44}g\frac{n^2}{r^2} + I_1\omega^2$$

$$L_{34} = KA_{55}g\frac{d}{dr} + KA_{55}g' + KA_{55}g\frac{1}{r}, \; L_{35} = KA_{44}g\frac{n}{r}, \; L_{41} = 2B_{16}g\frac{n}{r}\frac{d}{dr} + B_{16}g'\frac{n}{r}$$

$$L_{42} = B_{16}g\frac{d^2}{dr^2} - B_{26}g\frac{1}{r}\frac{d}{dr} + B_{16}g'\frac{d}{dr} - B_{16}g'\frac{1}{r} + B_{26}g\frac{1}{r^2} + B_{26}g\frac{n^2}{r^2}$$

$$L_{43} = -KA_{55}g\frac{d}{dr}$$

$$L_{44} = D_{11}g\frac{d^2}{dr^2} + D_{11}g'\frac{d}{dr} + D_{11}g\frac{1}{r}\frac{d}{dr} + D_{12}g'\frac{1}{r} - D_{22}g\frac{1}{r^2} + D_{66}g\frac{n^2}{r^2} - KA_{55}g + I_3\omega^2$$

$$L_{45} = D_{12}g\frac{n}{r}\frac{d}{dr} + D_{66}g\frac{n}{r}\frac{d}{dr} + D_{12}g'\frac{n}{r} - D_{22}g\frac{n}{r^2} - D_{66}g\frac{n}{r^2}$$

$$L_{51} = B_{16}g\frac{d^2}{dr^2} + B_{16}g'\frac{d}{dr} + 2B_{16}g\frac{1}{r}\frac{d}{dr} + B_{26}g\frac{1}{r}\frac{d}{dr} + B_{26}g'\frac{1}{r} + B_{26}g\frac{1}{r^2} + B_{26}g\frac{n^2}{r^2}$$

$$L_{52} = B_{26}g'\frac{n}{r} + 2B_{26}g\frac{n}{r}\frac{d}{dr}, \; L_{53} = -KA_{44}g\frac{n}{r}$$

$$L_{54} = D_{66}g\frac{n}{r}\frac{d}{dr} + D_{12}g\frac{n}{r}\frac{d}{dr} + D_{66}g'\frac{n}{r} + D_{66}g\frac{n}{r^2} + D_{22}g\frac{n}{r^2}$$

$$L_{55} = D_{66}g\frac{d^2}{dr^2} + D_{66}g\frac{1}{r}\frac{d}{dr} + D_{66}g'\frac{d}{dr} - KA_{44}g - D_{66}g'\frac{1}{r} - D_{66}g\frac{1}{r^2} + D_{22}g\frac{n^2}{r^2} + I_3\omega^2$$

Introducing the nondimensional parameters, $R = \frac{r-a}{l}$, $\quad a \le r \le b \quad and \ R \in [0,1]$, and $\lambda = \omega l \sqrt{\frac{I_1}{A_{11}}}$; a frequency parameter, $\beta = \frac{a}{b}$; the radii ratio, $\gamma = \frac{h}{r_a}$; the ratios of thickness to the radius of inner circle, $\delta_k = \frac{h_k}{h}$, and the relative layer thickness of the *k*-th layer. (10)

Here, $h_k$ is the thickness of the *k*-th layer, *h* is the total thickness of the plate, $r_a$ is the radius of inner circular plate and $A_{11}$ is a standard extensional rigidity coefficient.

After nondimensionalizing Equation (9), a new set of differential equations is obtained.

### 2.1. Thickness Variation

Case (i):

If $C_e = C_s = 0$, then the thickness variation becomes linear. In this case, it can easily be shown that $C_\ell = \frac{1}{\eta} - 1$, where $\eta$ is the taper ratio $h_k(0)/h_k(1)$.

Case (ii):

If $C_\ell = C_s = 0$, then the excess thickness over uniform thickness varies exponentially.

Case (iii):

If $C_\ell = C_e = 0$, then the excess thickness varies sinusoidally.

It may be noted that the thickness of any layer at the end $R = 0$ is $h_{0k}$ for Cases (i) and (iii) but is $h_{0k}(1 + C_e)$ for Case (ii).

The following range of values of the thickness coefficients is considered:

$$0.5 \le \eta \le 2.1, \ -0.2 \le C_e \le 0.2, \ -0.5 \le C_s \le 0.5$$

### 2.2. Method of Solution

The spline method is used because it uses a series of lower-order approximations rather than global higher-order approximations, affording fast convergence and high accuracy.

The differential equations consist of second-order derivatives in $U(R)$, $V(R)$, $W(R)$, $\Psi_r(R)$ and $\Psi_\theta(R)$. Therefore, these functions are approximated using cubic splines as:

$$
\begin{aligned}
U^*(R) &= \sum_{i=0}^{2} a_i R^i + \sum_{j=0}^{N-1} b_j (R - R_j)^3 H(R - R_j) \\
V^*(R) &= \sum_{i=0}^{2} c_i R^i + \sum_{j=0}^{N-1} d_j (R - R_j)^3 H(R - R_j) \\
W^*(R) &= \sum_{i=0}^{2} e_i R^i + \sum_{j=0}^{N-1} f_j (R - R_j)^3 H(R - R_j) \\
\Psi_r^*(R) &= \sum_{i=0}^{2} g_i R^i + \sum_{j=0}^{N-1} p_j (R - R_j)^3 H(R - R_j) \\
\Psi_\theta^*(R) &= \sum_{i=0}^{2} l_i R^i + \sum_{j=0}^{N-1} q_j (R - R_j)^3 H(R - R_j)
\end{aligned}
\tag{10}
$$

in which $H(R - R_j)$ is the Heaviside function and $a_i$, $c_i$, $e_i$, $g_i$, $l_i$, $b_j$, $d_j$, $f_j$, $p_j$ and $q_j$ are unknown coefficients (i.e., spline coefficients).

Let us assume that the interval $R \ \varepsilon \ [\ 0, \ 1\ ]$ is divided into *N* equal subintervals. The knots are at $R = R_s = \frac{s}{N}$; $s = 0, 1, \ldots, N$. We have the system of $(5N + 5)$ homogeneous equations in $(5N + 15)$ spline coefficients.

The simply supported (S-S) boundary condition is considered.

Each of these cases gives 10 more equations, thus making a total of $(5N + 15)$ equations, in the same number of unknowns. The resulting field and boundary condition equations may be written in the form

$$[M]\{q\} = \lambda^2 [P]\{q\} \tag{11}$$

where $[M]$ and $[P]$ are square matrices and $\{q\}$ is a column matrix. This is treated as a generalized eigenvalue problem in the eigenparameter $\lambda$ and the eigenvector $\{q\}$, whose elements are the spline coefficients.

## 3. Results and Discussion

In this work, free vibration of annular circular plates of variable thickness with anti-symmetric angle-ply lamination schemes with different material combinations is studied. Materials used in the analysis are Kevlar-49/epoxy (KGE), graphite/epoxy (AS4/3501-6) (AGE) and E-glass/epoxy (EGE). The spline method is used to approximate the displacement functions for analyzing the vibration behavior of the layered annular circular plates for S-S boundary conditions. In all the analysis, circumferential node number $n = 2$ is selected because frequency is lowest at this point, which shows that structure is least rigid and most flexible at this point.

*Validation*

Table 1 shows the comparison of a reduced case of current results with [24–26] for simply supported annular circular plates, which is the validation of the present results with the available results.

**Table 1.** Comparison of natural frequency parameter (rad/s) of present study $\lambda = \omega\ell\sqrt{\frac{I_0}{A_{11}}}$ with [24–26] of simply supported annular circular plates $\beta = 1/6 \ and \ \gamma = 1/60$.

| Circumferential Node Number $n$ | Modes | Present | Hashemi et al. (2010a) Exact Sol. | Duan et al. (2005) FSDT | Liu et al. (2008) FEM |
|---|---|---|---|---|---|
| 1 | 1 | 1583 | 1584.65 | 1583 | 1551 |
|   | 2 | 5436 | 5437.61 | 5433 | 5328 |
|   | 3 | 11,494 | 11,495.1 | 11,478 | 11,283 |
| 2 | 1 | 2307 | 2308.11 | 2306 | 2260 |
|   | 2 | 6474 | 6475.13 | 6468 | 6348 |
|   | 3 | 12,652 | 12,653.2 | 12,632 | 12,428 |

Tables 2–4 show the influence of taper ratio $\eta$, exponential coefficient of variation $C_e$ and sinusoidal coefficient of variation $C_s$ on the fundamental frequency parameter $\lambda$. Two- and four-layered annular circular plates are considered with circumferential node number $n = 2$, ratio of thickness to radius of inner circle $\gamma = 0.05$ and radii ratio $\beta = 0.5$. It can be seen that the value of the frequency parameter increases with the increase in number of layers. Frequency increases as number of layers increases, showing that the structure's rigidity increases and flexibility decreases. Similarly, the frequency parameter value increases with the increase in thickness coefficients, evidently showing that the rigidity of the structure increases and flexibility decreases.

**Table 2.** Relation of linear thickness variation $\eta$ and the fundamental frequency parameter $\lambda$. $n = 2$, $\gamma = 0.05$ and $\beta = 0.5$.

| | $\lambda$ | | |
|---|---|---|---|
| $\eta$ | $45^0/-30^0/30^0/-45^0$ (KGE/EGE/EGE/KGE) | $45^0/-60^0/60^0/-45^0$ (KGE/EGE/EGE/KGE) | $30^0/-30^0$ (EGE/EGE) |
| 0.5 | 0.71949 | 0.643041 | 0.402384 |
| 0.7 | 0.728127 | 0.672854 | 0.421916 |
| 0.9 | 0.734973 | 0.690198 | 0.447392 |
| 1.1 | 0.739494 | 0.700368 | 0.464501 |
| 1.3 | 0.741874 | 0.706052 | 0.476395 |
| 1.5 | 0.74287 | 0.709517 | 0.484848 |
| 1.7 | 0.742917 | 0.711673 | 0.490948 |
| 1.9 | 0.742298 | 0.711698 | 0.495378 |
| 2.1 | 0.741207 | 0.710538 | 0.498588 |

**Table 3.** Relation of exponential thickness variation $C_e$ and the fundamental frequency parameter $\lambda$. $n = 2$, $\gamma = 0.05$ and $\beta = 0.5$.

| | $\lambda$ | | |
|---|---|---|---|
| $C_e$ | $45^0/-30^0/30^0/-45^0$ (KGE/EGE/EGE/KGE) | $45^0/-60^0/60^0/-45^0$ (KGE/EGE/EGE/KGE) | $30^0/-30^0$ (EGE/EGE) |
| $-0.2$ | 0.745785 | 0.713475 | 0.490829 |
| $-0.1$ | 0.742443 | 0.70506 | 0.472436 |
| 0 | 0.737531 | 0.695956 | 0.456753 |
| 0.1 | 0.732985 | 0.687592 | 0.44392 |
| 0.2 | 0.729205 | 0.68026 | 0.433387 |

**Table 4.** Relation of sinusoidal thickness variation $C_s$ and the fundamental frequency parameter $\lambda$. $n = 2$, $\gamma = 0.05$ and $\beta = 0.5$.

| | $\lambda$ | | |
|---|---|---|---|
| $C_s$ | $45^0/-30^0/30^0/-45^0$ (KGE/EGE/EGE/KGE) | $45^0/-60^0/60^0/-45^0$ (KGE/EGE/EGE/KGE) | $30^0/-30^0$ (EGE/EGE) |
| $-0.5$ | 0.673905 | 0.637247 | 0.43449 |
| $-0.3$ | 0.711461 | 0.677643 | 0.452291 |
| $-0.1$ | 0.731285 | 0.692456 | 0.457029 |
| 0.1 | 0.74213 | 0.696611 | 0.455322 |
| 0.3 | 0.748382 | 0.693315 | 0.45001 |
| 0.5 | 0.751603 | 0.685991 | 0.442494 |

However, Table 5 depicts the comparison of four-layered annular circular plates, concluding that different lamination schemes and material combinations definitely affect the frequency of plates. It is evident that ply orientation $30^0/-30^0/30^0/-30^0$ shows the least frequency, proving that the structure is most flexible using this ply orientation, and $45^0/-45^0/45^0/-45^0$ shows the maximum frequency, which proves that the structure is least flexible using this ply orientation. Table 6 shows the elastic properties of materials used.

**Table 5.** Relation of linear thickness variation $\eta$ and the fundamental frequency parameter $\lambda$. $n = 2$, $\gamma = 0.05$ and $\beta = 0.5$.

| | $\lambda$ | | | |
|---|---|---|---|---|
| $\eta$ | $45^0/-30^0/30^0/-45^0$ (KGE/EGE/EGE/KGE) | $30^0/-30^0/30^0/-30^0$ (AGE/EGE/EGE/AGE) | $45^0/-60^0/60^0/-45^0$ (KGE/EGE/EGE/KGE) | $45^0/-45^0/45^0/-45^0$ (KGE/EGE/EGE/KGE) |
| 0.5 | 0.71949 | 0.487129 | 0.643041 | 0.84425 |
| 0.7 | 0.728127 | 0.503465 | 0.672854 | 0.848355 |
| 0.9 | 0.734973 | 0.51629 | 0.690198 | 0.854028 |
| 1.1 | 0.739494 | 0.525374 | 0.700368 | 0.855654 |
| 1.3 | 0.741874 | 0.531657 | 0.706052 | 0.855741 |
| 1.5 | 0.74287 | 0.535933 | 0.709517 | 0.854683 |
| 1.7 | 0.742917 | 0.538766 | 0.711673 | 0.852878 |
| 1.9 | 0.742298 | 0.540549 | 0.711698 | 0.850596 |
| 2.1 | 0.741207 | 0.54156 | 0.710538 | 0.848079 |

**Table 6.** Elastic properties of materials used.

| Elastic Property | Density $\times 10^3$ N$-$s$^2$/m$^4$ | Young Modulus $E_x \times 10^{10}$N/m$^2$ | Young Modulus $E_y \times 10^{10}$N/m$^2$ | Shear Modulus $G_{xz} \times 10^{10}$N/m$^2$ | Shear Modulus $G_{yz} \times 10^{10}$N/m$^2$ | Shear Modulus $G_{xy} \times 10^{10}$N/m$^2$ | Major Poisson Ratio, $\upsilon_{xy}$ |
|---|---|---|---|---|---|---|---|
| EGE | 1440 | 5.52 | 86.19 | 2.07 | 1.72 | 2.07 | 0.34 |
| AGE | 2550 | 11.72 | 42.75 | 4.14 | 3.45 | 4.14 | 0.27 |
| KGE | 1770 | 9.65 | 144.8 | 4.14 | 3.45 | 4.14 | 0.30 |

Figure 3 shows the variation of the fundamental frequency parameter with respect to the circumferential node number of two-layered annular circular plates for three thickness variations $C_s = 0.5$, $C_e = 0.2$ and $\eta = 1.5$ under S-S boundary conditions. It can be seen that at circumferential node number $n = 2$ frequency is lowest, which shows that the structure is least rigid and most flexible at this point. As circumferential node number increases, frequency value also increases, which shows that rigidity increases but flexibility decreases.

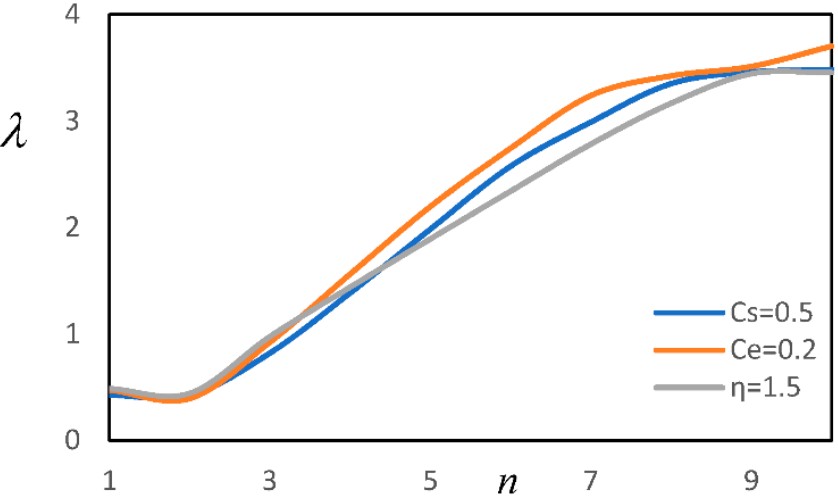

**Figure 3.** The relation of fundamental frequency parameter and circumferential node number of two-layered annular circular plates with ply orientation $30^0/-30^0$ ($KGE/KGE$). $\beta = 0.5$ and $\gamma = 0.05$.

Figure 4 narrates the angular frequency parameter $\omega$ with respect to radii ratio $\beta$ for two-layered annular circular plates with different materials. It can be seen that angular frequency increases gradually from $\beta = 0.1$ *to* 0.5 but strictly increases afterwards. It shows that with the increase in $\beta$, rigidity increases but flexibility decreases. Figure 5 shows the trend of the angular frequency parameter and radii ratio of four-layered annular plates after fixing thickness variations $C_e = 0.2$, $C_s = 0.5$ and $\eta = 1.5$. Moreover, a relationship between the angular frequency parameter and the radii ratio of four-layered annular plates after fixing thickness variation $C_e = 0.2$ can be seen in Figure 6. However, Figure 7 relates the angular frequency parameter and the radii ratio of four-layered annular plates with different lamination schemes and lamination materials, concluding that both of these factors definitely affect the angular frequency parameter value. The first three mode shapes of vibration of annular circular plates are presented in Figure 8a–c. The shape is not circular because only a small portion of the plate is considered so that the mode shapes can be visualized clearly. Normalization is conducted with respect to the maximum transverse displacement W. As expected, the transverse displacements are most dominant.

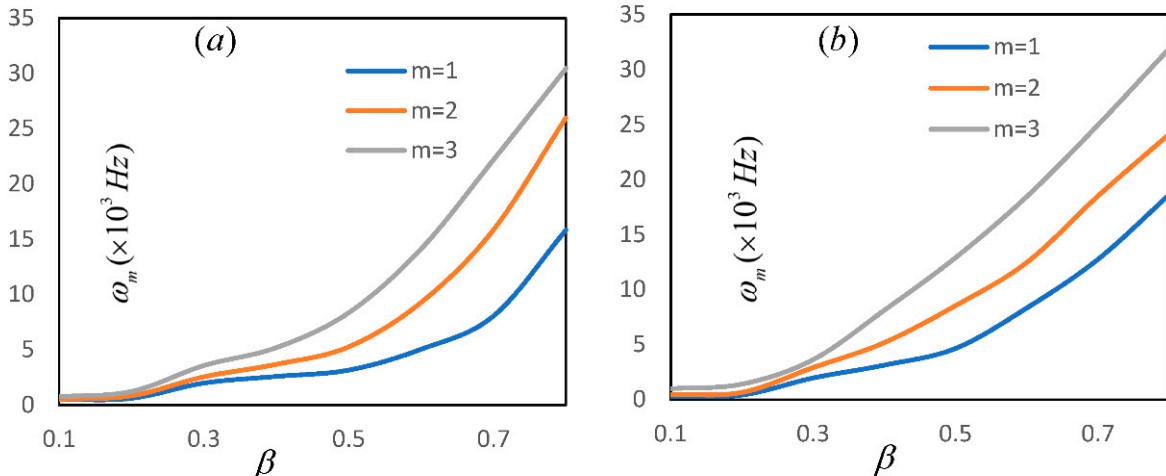

**Figure 4.** The relation of angular frequency parameter and radii ratio $\beta$ of two-layered annular circular plates with ply orientation $45^0/-45^0$: (**a**) $EGE/EGE$ (**b**) $AGE/AGE$. $\eta = 1.5$, $n = 2$ and $\gamma = 0.05$.

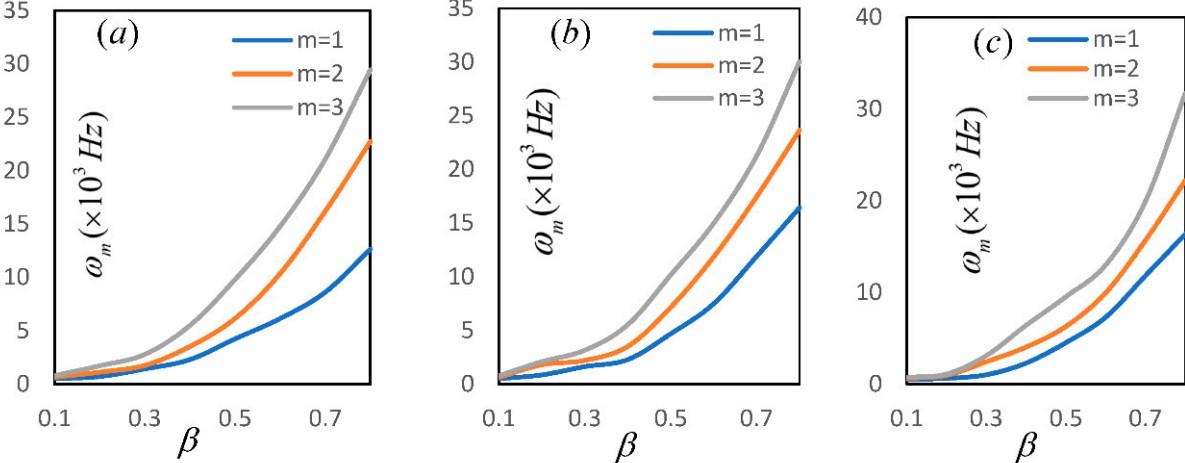

**Figure 5.** The relation of angular frequency parameter and radii ratio of four-layered annular circular plates with ply orientation $45^0/-30^0/30^0/-45^0$ ($AGE/EGE/EGE/AGE$): (**a**) $C_e = 0.2$; (**b**) $C_s = 0.5$; (**c**) $\eta = 1.5$; $n = 2$ and $\gamma = 0.05$.

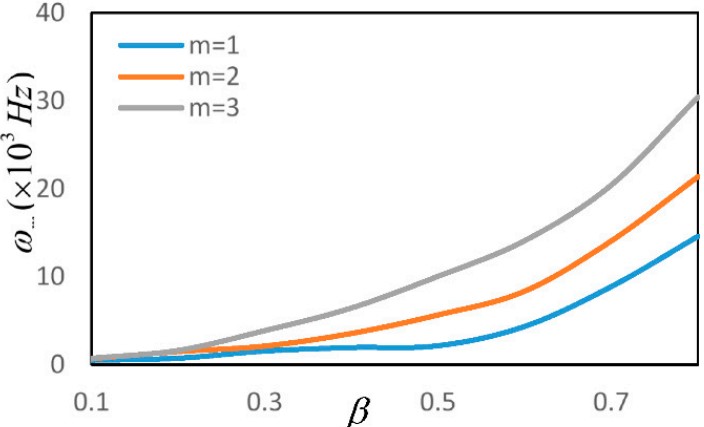

**Figure 6.** The relation of radii ratio and the angular frequency parameter of four-layered annular. circular plates with ply orientation $45^°/-30^°/30^°/45^°$ ($KGE/EGE/EGE/KGE$). $C_e = 0.2$, $n = 2$ and $\gamma = 0.05$.

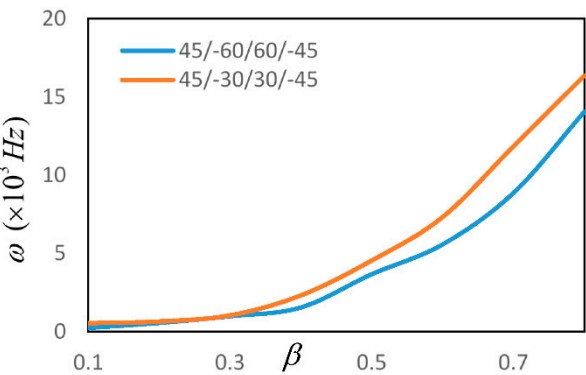

**Figure 7.** The relation of radii ratio and fundamental angular frequency parameter of four-layered annular circular plates with ply orientation $45^{\circ}/-30^{\circ}/30^{\circ}/45^{\circ}$ $(AGE/EGE/EGE/AGE)$. $\eta = 1.5$, $n = 2$ and $\gamma = 0.05$.

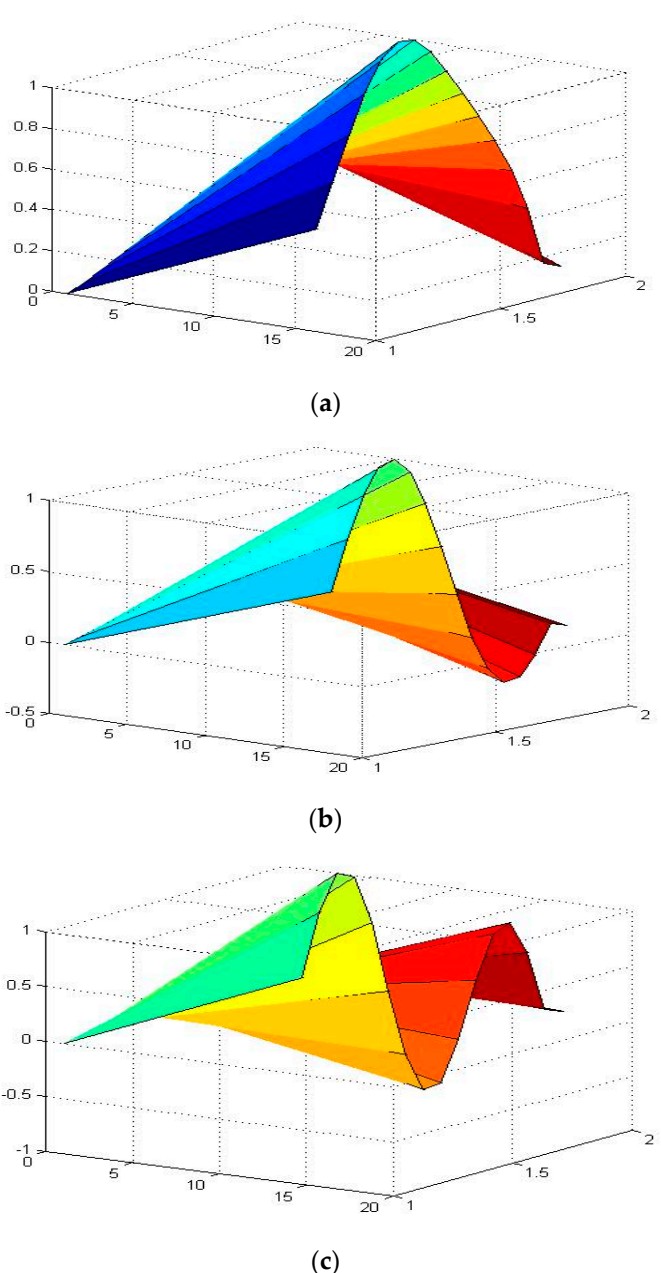

**Figure 8.** (**a–c**) Mode shapes of annular circular plate (W).

## 4. Conclusions

The present study investigates the free vibration of antisymmetric angle-ply annular circular plates having variable thickness under shear deformation theory for simply supported boundary conditions. The vibration behavior of the annular plates is examined for circumferential node number, radii ratio, thickness variations, number of layers and stacking sequences. Frequency parameter value increases with the increase in all thickness coefficients; radii ratio and number of layers evidently show that rigidity of the structure increases and flexibility decreases. The relation between circumferential node number and frequency parameter value shows that at circumferential node number $n = 2$ frequency is lowest, which shows that the structure is least rigid and most flexible at this point. As circumferential node number increases, frequency value also increases, which shows that rigidity increases but the flexibility decreases. If the boundary conditions are altered to clamped, free or any combination of these two, we can obtain different results, but they will be closer to the current results. Hence, it is concluded that variation of the geometric and material parameters affect the frequency of the considered annular circular plate.

**Funding:** This research was funded by [The Deanship of Scientific Research at King Faisal University, Saudi Arabia] grant number [Nasher Track (Grant No. NA00037)] And The APC was funded by [The Deanship of Scientific Research at King Faisal University, Saudi Arabia].

**Institutional Review Board Statement:** Not applicable.

**Informed Consent Statement:** Not applicable.

**Acknowledgments:** This work was completed by Saira Javed. The author gratefully acknowledges the Deanship of Scientific Research at King Faisal University, Saudi Arabia, for the financial support under Nasher Track (Grant No. NA00037).

**Conflicts of Interest:** The authors declare no conflict of interest.

## Nomenclature

| | |
|---|---|
| EGE | E-glass/epoxy |
| GE | AS4/3501-6 graphite/epoxy |
| KE | Kevlar-49/epoxy |
| S−S | Both the ends are simply supported |
| Symbols | |
| $A_{ij}$ | Elastic coefficients representing the extensional rigidity |
| $A_{ij}^c$ | Variable thickness elastic coefficients representing the extensional rigidity of uniform thickness |
| $B_{ij}$ | Elastic coefficients representing the bending-stretching coupling rigidity |
| $B_{ij}^c$ | Variable thickness elastic coefficients representing the bending-stretching coupling rigidity of uniform thickness |
| $C_e$, $C_\ell$, $C_s$ | Exponential, linear and sinusoidal variation, respectively |
| $D_{ij}$ | Elastic coefficients representing the bending rigidity |
| $D_{ij}^c$ | Variable thickness elastic coefficients representing the bending rigidity of uniform thickness |
| $H$ | Side-to-thickness ratio |
| $H(R - R_j)$ | The Heaviside function |
| $I_1$ | Normal inertia coefficient |
| $I_3$ | Inertia coefficients |
| $K$ | Shear correction factor |
| $L$ | Length parameter |
| $L_{ij}$, $L_{ij}^*$ | Differential operator occurring in the equations of motion |
| $\left.\begin{array}{l} M_r \\ M_\theta \\ M_{r\theta} \end{array}\right\}$ | Moment resultants in the respective direction of annular circular plate |

| | |
|---|---|
| $\left.\begin{array}{l} N_r \\ N_\theta \\ N_{r\theta} \end{array}\right\}$ | Stress resultants in the respective direction of annular circular plate |
| $N$ | Number of intervals of spline interpolation |
| $Q_{ij}^{(k)}$ | Elements of the stiffness matrix for the material of $k$-th layer |
| $\overline{Q}_{ij}^{(k)}$ | Elements of the transformed stiffness matrix for the material of $k$-th layer |
| $\left.\begin{array}{l} Q_{rz} \\ Q_{\theta z} \end{array}\right\}$ | Transverse shear resultants in the respective directions |
| $R$ | Radial distance coordinate |
| $U, \; V, \; W$ | Displacement functions in $r, \theta, z$ directions |
| $\overline{U}, \; \overline{V}, \; \overline{W}$ | Nondimensionalized displacement functions in $r, \theta, z$ directions |
| $R_s$ | The equally spaced knots of spline interpolation |
| $a, b$ | Length and width of the inner radius and the outer radius |
| $\left.\begin{array}{l} a_i \\ c_i \\ e_i \\ g_i \\ l_i \end{array}\right\},$ | |
| $\left.\begin{array}{l} b_j \\ d_j \\ f_j \\ p_j \\ q_j \end{array}\right\}$ | Spline coefficients |
| $h$ | Total thickness |
| $h_k$ | Thickness of the $k$-th layer |
| $i, j, k$ | General indices |
| $\ell$ | Width of annular circular plate |
| $n$ | Circumferential node number |
| $r$ | Radius of reference surface of plate at a general point |
| $r_a, \; r_b$ | The radius of the inner radius and the outer radius |
| $u, \; v, \; w$ | $r, \theta, z$ displacements |
| $u_0, \; v_0$ | The in-plane displacements of the reference surface |
| $z$ | Normal coordinate of any point on the annular circular plate |
| $z_k$ | Distance to the top of the $k$-th layer from the reference surface |
| $\beta$ | The radii ratio $a/b$ |
| $\delta_k$ | Relative layer thickness of the $k$-th layer |
| $\left.\begin{array}{l} \varepsilon_x \\ \varepsilon_y \\ \varepsilon_\theta \\ \varepsilon_r \end{array}\right\}$ | Normal strain in the respective directions |
| $\gamma$ | Ratios of thickness to radius of inner circle |
| $\left.\begin{array}{l} \gamma_{r\theta} \\ \gamma_{rz} \\ \gamma_{\theta z} \end{array}\right\}$ | Shear strain in the respective directions |
| $\omega$ | Angular frequency |
| $\psi_r, \; \psi_\theta$ | Shear rotations of any point on the middle surface |
| $\Psi_r, \; \Psi_\theta$ | Shear rotational functions |
| $\overline{\Psi}_R, \; \overline{\Psi}_\Theta$ | Nondimensionalized shear rotations |
| $\rho$ | Mass density of the material |
| $\left.\begin{array}{l} \sigma_r \\ \sigma_\theta \end{array}\right\}$ | Normal stress in the respective directions |
| $\left.\begin{array}{l} \tau_{r\theta} \\ \tau_{\theta z} \\ \tau_{rz} \end{array}\right\}$ | Shear stress at a point on the reference surface |
| $\theta$ | Ply orientation angle |

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
