# Peer review of "Natural Frequencies Calculation of Composite Annular Circular Plates with Variable Thickness Using the Spline Method"

_jcs, doi:10.3390/jcs6030070_

Round 1

Reviewer 1 Report

Presented problem is scientifically interesting and has the utilitarian meaning. However, some elements should be improved:

  • the beginning of sentence “[8] analyses …” in Introduction point,
  • the new elements , the novelty of undertaken investigations should be stronger underlined,
  • the function (4) expressing the plate thickness variation could be presented graphically with the reference to the source work,
  • the notation of the thic-ness should be improved,
  • there are the same mark of parameters Aijc including the shear correction and not including, so may be notation with the different marks will be better,
  • the examined object should be defined more clearly with the scheme presentation. Some confusion with parameters a and ra
  • in Results and discussion point the explanation connected with choose of the number n=2 of circumferential node should be presented,
  • both points of work: result analysis and conclusions should be presented more insightful,
  • the abbreviation (S-S) boundary can be used after the description, which is presented in the next point: Results and discussion.

Author Response

Respected Dr.,

Greetings,

I have completed the revision as per your instructions and point -by-point reply is attached.

Reviewer 2 Report

The topic addressed by the author is relevant, but the manuscript needs to be improved.

The introductory section where the state of the art is presented needs a significant improvement, both in terms of the references considered as well as concerning the discussion of those studies, those conclusions and insights to further studies. This section should provide the support and justify the need for the work presented.

It is recommended to include figures to better illustrate the references to the annular plates characteristics mentioned in the text in section 2, its relation to the coordinate system and the effects the C parameters have in the thickness description among other aspects.

Please include the meaning of the C parameters in equation 4.

It is recommended in section 2 to illustrate the discretization using spline functions.

It is recommended to clarify the achievement of equilibrium equation (in eq. 12) and their meaning. The manuscript must consider the physical problem to solve and not just the mathematical steps needed.

The manuscript does not consider any validation or verification case studies. This is a fundamental preliminary aspect, that should be considered.

The discussion of results should not be restricted to the description of what can be observed in the tables and figures. It is highly recommended to provide a physical meaning to the results and their relation to the parameters analysed. 

Conclusions must be more objective, besides clarifying which shear deformation theory it was used, the influence of the different parameters should be considered in more detail, namely by quantifying the more advantageous solutions and why, for example.

Author Response

(The authors gave the same response as above.)

Reviewer 3 Report

The author presents a method to calculate natural frequencies of variable thickness composite annular circular plates. This is dine using the spline method.

The manuscript carries the title of "Spline method analysis for composite annular circular plates with varying thickness" and it is misleading.

It should have the word "vibration" so I suggest to modify the title in something like:

"Calculation of natural frequencies of variable thickness composite annular circular plates using the spline method"

Coming to the content of the paper:

The topic of the paper had been investigated in depth. So what is the novelty of the present paper ? Please explain and discuss in the manuscript.

The numerical results obtained using the spline method are not compared with existing results from the literature or with FE calculation. This is crucial to assess the reliability of the presented results.

Moreover, once the natural frequencies are calculated, their associate mode shapes should be presented. This is missing in the present paper.

How is Eq. 12 solved ? Please discuss the accuracy of the calculations.

The author assumes an "easy" boundary condition: Simply supported at both circumferences. What would happen, if the boundary conditions are altered to clamped, free and any combination of these two. Please discuss this issue.

Author Response

(The authors gave the same response as above.)

Round 2

Reviewer 2 Report

The majority of the comments made, were addressed in an acceptable manner. However, the introductory section do needs an improvement. It is not simply a matter of including references, which was done, but it is related with the way the work already published by other researchers is described, and the need to evidence that your work goes beyond other published work.

Author Response

Respected Reviewer,

Greeting.

Modified manuscript as per your instructions is attached.

Reviewer 3 Report

The author provided adequate answer to most of my comments, however some of them remained without a good reply:

Point 1: The title of the revised manuscript was modified to :

"Natural frequency calculation of composite annular circular plates with variable thickness using the spline method"

however as the calculation provides more than one natural frequency for the annular circular plates, the second word in the title should be " frequencies".

Point 2:  Claiming that :"The novelty of current research is that spline method is not be used by any of above-mentioned researchers to analyze free vibration of annular circular plates of variable thickness. Moreover, use of antisymmetric layers are also novelty of the current research" is too simply. Browsing the web yield many of paper using the spline method, like the typical 3 ones:

https://doi.org/10.1155/2005/128640, 

10.2140/jomms.2008.3.929,

https://doi.org/10.1515/jaa-2021-2078

Therefore a better discussion should be used for this point.

Point 4: The modes presented in Fig. 8 are peculiar : Is it the first three modes of the annular circular plates ? Then write it down in the text. Why the shape is not circular ? Please address this issue in depth.

Point 5: The explanation written to me for this point should appear in the text of the paper.

Finally an important issue which is not clearly presented in the manuscript:

The properties of the composite layers are given in a x,y,z system of coordinates, while the problem is solved using a polar system of coordinates, r and theta. Therefore how are the material constants fitted to the polar solution ?

Moreover, the properties of the materials used for the calculations should appear in a Table.

Author Response

(The authors gave the same response as above.)

Round 3

Reviewer 3 Report

The author provided adequate answers and modification to the manuscript and improved greatly its content.

The modifications were not signed in yellow, however I could see that all the relevant additions were in the third version of the paper.